molecular biology/genetics/ecology

chemical communication, MHC, microbiome, preen oil, uropygial gland secretion

**Author for correspondence:**
L. A. Grieves
e-mails: grievel@mcmaster.ca, lgrieves@uwo.ca

†Present address: McMaster University, Department of Psychology, Neuroscience & Behaviour, 1280 Main St. W, Hamilton, Ontario, Canada, L8S 4L8.

# Preen gland microbiota covary with major histocompatibility complex genotype in a songbird

L. A. Grieves[1,†], G. B. Gloor[2], M. A. Bernards[1] and E. A. MacDougall-Shackleton[1]

[1]Department of Biology, University of Western Ontario, London, ON, Canada N6A 5B7
[2]Department of Biochemistry, University of Western Ontario, London, ON, Canada N6A 5C1

LAG, 0000-0002-6836-2177; EAM-S, 0000-0002-4138-6398

Pathogen-mediated selection at the major histocompatibility complex (MHC) is thought to promote MHC-based mate choice in vertebrates. Mounting evidence implicates odour in conveying MHC genotype, but the underlying mechanisms remain uncertain. MHC effects on odour may be mediated by odour-producing symbiotic microbes whose community structure is shaped by MHC genotype. In birds, preen oil is a primary source of body odour and similarity at MHC predicts similarity in preen oil composition. Hypothesizing that this relationship is mediated by symbiotic microbes, we characterized MHC genotype, preen gland microbial communities and preen oil chemistry of song sparrows (*Melospiza melodia*). Consistent with the microbial mediation hypothesis, pairwise similarity at MHC predicted similarity in preen gland microbiota. Counter to this hypothesis, overall microbial similarity did not predict chemical similarity of preen oil. However, permutation testing identified a maximally predictive set of microbial taxa that best reflect MHC genotype, and another set of taxa that best predict preen oil chemical composition. The relative strengths of relationships between MHC and microbes, microbes and preen oil, and MHC and preen oil suggest that MHC may affect host odour both directly and indirectly. Thus, birds may assess MHC genotypes based on both host-associated and microbially mediated odours.

## 1. Introduction

The major histocompatibility complex (MHC), present in all jawed vertebrates, is a highly polymorphic gene family. MHC gene products are receptors that recognize invading antigens, then bind and present them to T cells to initiate an adaptive immune response [1]. Pathogen-mediated selection often favours

particular allelic combinations; in some systems, there is an apparent preference for maximally MHC-diverse individuals (e.g. in fat-tailed dwarf lemurs, *Cheirogaleus medius* [2]) while in others, intermediate or optimal MHC diversity is preferred (e.g. in three-spined stickleback, *Gasterosteus aculeatus* [3,4]). In general, individuals with a greater variety of alleles have broader antigen-binding repertoires and should be better protected against infectious disease [5]. These fitness effects may promote sexual selection in the form of MHC-based mate choice, and indeed mating preferences for individuals with particular MHC profiles have been documented across vertebrate taxa [6]. Accordingly, mechanisms must exist for assessing the MHC diversity or similarity of potential mates.

A growing body of evidence implicates olfaction as being central to assessing MHC profiles [3,7,8]. However, the mechanisms linking MHC to odour are not fully understood. An individual's MHC genotype may influence its odour relatively directly, if MHC molecules and/or the antigens that bind to them are odorous and excreted through urine or sweat [9,10]. An alternative route, not mutually exclusive, is for MHC genotype to influence odour indirectly, mediated through the community composition of odour-producing symbiotic bacteria [9,10]. This possibility is most feasible when considering MHC class II genes, which interact with extracellular microbes such as bacteria [1,11]. MHC genes may alter microbial community composition via bacterial surface molecules that bind to specific host cell proteins (where bacteria that cannot adhere to host cells are shed) and/or through antigen-mediated elimination of select bacterial species [12,13]. Thus, an individual's genotype at MHC class II may shape the community composition of symbiotic bacteria that are able to survive on or inside the host, and the resultant microbial communities may affect host odour in turn [14,15]. While the influence of MHC genotype on host microbial communities has received growing attention in recent years [16–18], this has rarely been explored in birds. However, in seabirds, MHC class II genotype has recently been correlated with microbiota of the feathers [19] and preen gland [20].

Although songbirds (order Passeriformes) were long thought to lack a robust olfactory system, we now know that this group perceives and uses olfactory information in a variety of contexts, including mate choice [21]. Song sparrows (*Melospiza melodia*) prefer the preen oil odour of MHC-dissimilar and MHC-diverse potential mates [8], and pairs of individuals that are more similar at MHC class II also have more similar preen oil chemistry [22]. These patterns raise the question of whether the observed link between MHC genotype and preen oil chemistry (a proxy for odour) stems primarily from direct or indirect (microbially mediated) influences.

The avian preen gland harbours diverse microbial communities (reviewed in [23]). Experimental evidence suggests that preen gland microbes produce volatile, preen oil-derived compounds associated with avian reproductive success [24]. While these volatile compounds are postulated to be associated with MHC genotype, this has not yet been investigated [24]. Here, we hypothesize that MHC class II genotype influences songbird odour primarily indirectly, mediated by the community composition of symbiotic microbes associated with the preen gland. If variation at MHC drives variation in preen gland microbial communities, and these microbes drive variation in preen oil composition, we predict that pairwise similarity at MHC should be significantly correlated with microbial similarity and microbial similarity should be significantly correlated with preen oil composition. Alternatively, if the effect of MHC genotype on odour is primarily direct, rather than mediated through microbial community composition, the strongest association should be that between MHC similarity and preen oil similarity. Recognizing that some but not all microbes might contribute to host odour, we used permutation as a complementary approach to identify subsets of microbes that may be particularly sensitive to MHC genotype or predictive of preen oil chemistry. We found support for both direct and indirect influences of MHC genotype on host odour. This suggests that, in assessing MHC profiles, birds may have access to both host-associated (direct) and microbially mediated (indirect) olfactory cues.

# 2. Methods

## 2.1. Sample collection

As part of previous studies, we characterized preen gland bacteria [23], conducted chemical analysis of preen oil [25,26] and genotyped the MHC class II of free-living adult song sparrows [8]. The present study provides a new analysis of these data and includes unpublished MHC class II data for one of our two sampling locations (Cambridge, see below).

We used seed-baited Potter traps and mist nets to capture birds at two locations in southern Ontario, Canada: on Western University property in London (43.008° N, 81.291° W) and at the *rare* Charitable Research Reserve in Cambridge (43.383° N, 80.357° W). We captured and sampled 31 adult song sparrows: 19 (11 males, 8 females) in London from 8 August to 1 September 2017 and 12 (8 males, 4 females) in Cambridge from 3 April to 1 May 2017.

From each bird, we gently probed the preen gland with an unheparinized capillary tube to express approximately 1–5 mg of preen oil. Immediately following preen oil collection, we swabbed the preen gland to collect microbes from both inside and outside the gland (see electronic supplementary material for details). The small body size of song sparrows (approx. 20 g) and the correspondingly small size of the preen gland and papilla (gland dimensions average 7.4 mm *W* by 4.2 mm *L*, papilla width averages 1.4 mm, electronic supplementary material, figure S1) prevent noninvasively and nonsurgically sampling microbes from directly within the gland while excluding microbes immediately outside the gland. Instead, because preen oil is frequently excreted from the preen gland, our external swabbing method was designed to collect bacteria living immediately outside the preen gland, as well as those inhabiting the gland. Because preen oil is routinely groomed through the feathers after being released from the preen gland, we think it likely that both bacteria inhabiting the preen gland and those around its external surface may modify the composition of preen oil on feathers through manipulation or degradation.

We collected a small blood sample (approx. 20 µl) from each bird through brachial venipuncture and, after the field season, sexed all birds using the P2/P8 PCR protocol described by Griffiths *et al*. [27]. This blood sample was also used for MHC genotyping. To ensure individuals were only sampled once, each bird was banded before release at the site of capture.

## 2.2. Bacterial DNA extraction and 16S amplification

Following [23], we extracted bacterial DNA from swabs using Qiagen DNeasy PowerSoil DNA isolation kits with some modifications to the manufacturer's recommended protocol (see electronic supplementary material for detailed protocol). Extractions were carried out in batches, each consisting of 23 preen gland swabs plus one extraction from a fresh sterile swab that served as a negative control. We amplified the V4 region of the bacterial 16S rRNA gene using the universal primers F518 [28] and R806 [29]. In addition to the priming sequences, each primer included an Illumina MiSeq adaptor sequence, four randomized nucleotides and a unique 'barcode' of eight nucleotides. We performed PCR in a total volume of 25 µl, including 10 µl of Quantabio 5PRIME HotMasterMix, 0.2 µM of each primer and 2 µl of DNA template (mean concentration = 0.1 ng µl$^{-1}$). The thermocycling profile consisted of 2 min at 94°C; 35 cycles of 45 s at 94°C, 60 s at 50°C and 90 s at 72°C; and a 10 min final extension at 72°C. Amplification was confirmed by running samples on a 2% agarose gel.

## 2.3. Sequencing and pipeline

We pooled PCR products into a library and sequenced with 250 nt paired-end reads on an Illumina MiSeq at the London Regional Genomics Centre. We used a pipeline [30] to collapse sequences into clusters of identical reads and assign sequences to individuals. We used a second pipeline [31] and the R package dada2 [32] to overlap reads, remove ambiguous reads and filter out chimaeras and singleton sequences (i.e. those appearing only once in the dataset) and those rarer than 0.1% in any sample. We assigned each unique sequence (i.e. sequence variants; hereafter SVs) to bacterial taxon by clustering at greater than or equal to 97% sequence identity (following [30]) using the naive Bayesian Ribosomal Database Project (RDP) Classifier [32,33]. The workflow and parameters used are the same as in [23] and are available at https://github.com/ggloor/miseq_bin/. Raw read count data, including taxonomic assignments for each sample, are available in the electronic supplementary material.

We used a compositional data analysis approach [34] that examines the read ratios between sequences. Most datasets do not actually contain all possible components; often, small values, including values below the detection limit of an instrument such as the Illumina MiSeq, are rounded off to zero. In such cases, zero counts reflect sampling or equipment limitations; that is, they are not true zeros but low counts for which the true value is unknown [35]. Because replacing these values with zero or discarding them can lead to estimation bias, such values should be imputed using an estimation method [35]. Accordingly, following [31], we used Bayesian-multiplicative replacement to impute values for zero count sequences using the R package zCompositions [35]. We then applied a

centred log-ratio (clr) transformation to the zero-replaced dataset, which renders the use of Euclidean distances meaningful in subsequent analyses [31,36].

Next, because rare sequences that occur in only a few samples are generally uninformative, and because samples with very low read counts are more likely to represent undersampling, we filtered sequences by the minimum proportion, minimum occurrence and minimum sample count of reads. Thus, sequences found in fewer than 0.5% of reads, sequences found in fewer than 10% of samples and samples with fewer than 5000 reads were removed from the initial dataset. We also removed three non-bacterial SVs identified as chloroplasts. The filtered dataset contained 44 SVs from across the 31 samples (mean ± s.e. SVs per individual = 26.4 ± 0.66, electronic supplementary material, table S1). Most of these retained SVs (39/44) were classified to the genus level. These methods were applied to a larger dataset [23]; none of the 31 samples used in this study were removed as a result of these processing steps. Detailed methods and quality control procedures are outlined in the electronic supplementary material.

## 2.4. MHC genetic analysis

We amplified the hypervariable second exon of MHC class II (approx. 350 nt) using primers SospMHCint1f [22] and Int2r.1 [37]. In addition to the priming sequences, each primer included an Illumina MiSeq adaptor sequence, four randomized nucleotides and a unique 'barcode' of eight nucleotides. Detailed PCR conditions and MHC sequencing methods are described elsewhere [8]. Briefly, we aligned amino acid sequences in MEGA v. 7.0 [38] and trimmed based on comparison to conspecific sequences in GenBank [39]. Trimming resulted in alleles of 73–86 amino acids, corresponding to most of exon II. Next, we used a pipeline [30] to collapse sequences into clusters of identical reads and assign sequences to individuals, retaining sequences comprising at least 1% of an individual's reads (mean ± s.e. retained reads per individual = 20 736 ± 1939).

We assigned each retained sequence to its corresponding protein sequence, removed any putative pseudogenes following [22], and applied Bayesian-multiplicative replacement and clr-transformation to the data to allow comparison to the microbial dataset. In some cases, different DNA sequence reads translated to the same amino acid sequence. For these, we calculated the average log-ratio value so that only unique protein sequences were included in further analysis. Across all 31 birds, we detected 151 unique amino acid alleles (mean ± s.e. alleles per individual = 16.23 ± 0.61).

## 2.5. Preen oil chemical analysis

We dissolved preen oil samples in 1–5 ml chloroform ($CHCl_3$; scaled for the volume of preen oil collected for a final concentration of 1 mg preen oil/ml $CHCl_3$) and analysed them using an Agilent 7890A gas chromatograph with flame ionization detector (GC-FID), fitted with a 5% phenyl methyl siloxane column (Agilent Technologies DB-5, 30 m × 0.32 µm ID × 0.25 µm film thickness). We followed the protocol and temperature profile described in [25]. Because the volume of preen oil collected varied across individuals, we quantified peak sizes based on the proportional peak size relative to total chromatogram peak area. To maintain comparability with the 16S and MHC genetic datasets we applied Bayesian-multiplicative replacement and clr-transformation to these proportional data. Across all 31 birds, we detected 65 unique preen oil peaks (mean ± s.e. peaks per individual = 24.6 ± 1.1).

## 2.6. Data analysis

All statistical analyses were performed in R v. 3.3.3 [40]. We used the clr data to construct Euclidean distance matrices for each dataset (preen gland microbial community composition, MHC amino acid genotype and preen oil chemical composition). Distances were calculated between all pairwise dyads. We compared these pairwise distance matrices (31 × 31 matrices, 961 pairwise combinations per matrix) in three separate tests. We ran Mantel tests in the vegan package [41] with 10 000 permutations to assess correlations (Spearman's $r$) between (i) MHC amino acid distance (hereafter 'MHC distance') and preen gland microbial distance (hereafter 'microbial distance'), (ii) microbial distance and preen oil chemical distance (hereafter 'chemical distance') and (iii) MHC distance and chemical distance.

Because we sampled birds from two locations and both sexes, for each of the MHC, microbial and chemical Euclidean distance matrices we performed permutational multivariate analysis of variance (PERMANOVA) tests to compare the magnitude of pairwise differences for individuals from different

populations and sexes. These analyses were done to assess the degree to which population or sex differences at MHC, microbial composition or preen oil chemistry might contribute to associations observed. This was particularly important because we pooled individuals from different populations and sexes and those sampled at different times of year. Pooling was done to maximize statistical power when testing for relationships between MHC, microbial and chemical distance.

Given the possibility that only a subset of microbes covary with MHC and/or influence preen oil composition, and that only a subset of preen oil chemicals reflect MHC, we used the 'bioenv' [42] function in the vegan package to identify the subset of bacterial SVs that best reflect MHC distance (which we interpreted as candidate SVs that may be particularly sensitive to MHC genotype), the subset of bacterial SVs that best predict chemical distance (interpreted as candidate SVs that may influence preen oil chemical composition) and the subset of preen oil chemical peaks that best reflect MHC distance (interpreted as candidate compounds for chemical cues of MHC genotype). This was achieved by considering all possible combinations of variables (SVs, chemical peaks) at increasing levels of complexity, up to a user-specified maximum (here, 6 SVs or peaks). The approach then identifies the variables that maximize the rank correlation between two distance matrices (here, MHC and microbial distance; microbial and chemical distance; MHC and chemical distance).

The rank correlations performed by bioenv are based on a large number of interdependent similarity calculations, so statistical tests of the bioenv results are inappropriate. Instead, these results are used as an exploratory tool, where the optimal variables identified are considered promising candidates for further study. Bias towards a more complicated explanation than the data support is not a major concern with this analysis, because the correlation coefficient tends to decrease with the inclusion of unimportant variables [42]. By selecting only the top six variables (out of a possible 44 SVs, 65 peaks, and 151 alleles) that maximize correlations, such bias is unlikely to be an issue in this dataset.

Finally, song sparrow preen oil is made up of wax monoesters arranged in a series of different combinations of different chain length fatty alcohols and fatty acids [25]. To infer the major and minor acid : alcohol esters in the subset of preen oil peaks that best reflect MHC distance in our bioenv analysis, we compared our GC-FID data with a previously published analysis of song sparrow preen oil that used GC-FID and GC-mass spectrometry approaches [25].

# 3. Results

Pairwise microbial distance was significantly positively correlated with pairwise MHC distance (Mantel test, Spearman's $r_{465} = 0.20$, $p = 0.014$; figure 1a) but not with pairwise chemical distance (Mantel test, Spearman's $r_{465} = 0.02$, $p = 0.389$; figure 1b). Pairwise MHC distance was significantly positively correlated with chemical distance (Mantel test, Spearman's $r_{465} = 0.39$, $p = 0.0007$; figure 1c), as has been found previously [8,22].

PERMANOVA on the Euclidian distance matrices for microbes, MHC and preen oil identified significant differences in preen gland microbiota, MHC genotype and preen oil chemistry among populations, but not between the sexes (table 1).

Permutation analysis (bioenv) identified a best combination of 6 SVs at which microbial and MHC distance matrices were maximally correlated (Mantel's $r = 0.42$, figure 2a, electronic supplementary material, table S2). Two of these SVs (*Micrococcus* and *Bacillus*) best maximized correlations (indicated by a change in Mantel's $r \geq 0.05$, electronic supplementary material, table S2), suggesting that the relative abundance of these taxa may be particularly sensitive to MHC genotype. We also identified a best combination of six SVs at which microbial and chemical distance matrices were maximally correlated (Mantel's $r = 0.48$, figure 2b, electronic supplementary material, table S3). Three of these SVs (*Methylobacterium*, *Bradyrhizobium* and *Xylophilus*) best maximized correlations, suggesting that these taxa are candidates for influencing preen oil chemical composition. We then identified a best combination of six GC-FID peaks at which chemical and MHC distance matrices were maximally correlated (Mantel's $r = 0.54$, figure 2c, electronic supplementary material, table S4). Two of these peaks (2, 24) best maximized correlations between preen oil and MHC, suggesting that they are candidates for chemical cues of MHC genotype. We were not able to infer the chemical composition of these peaks, but we inferred the major and minor wax monoesters of four of the six preen oil peaks that occurred repeatedly in all subset sizes (10, 18, 27 and 92) and an additional peak (51) that was identified in models with a smaller number of subsets (electronic supplementary material, table S5).

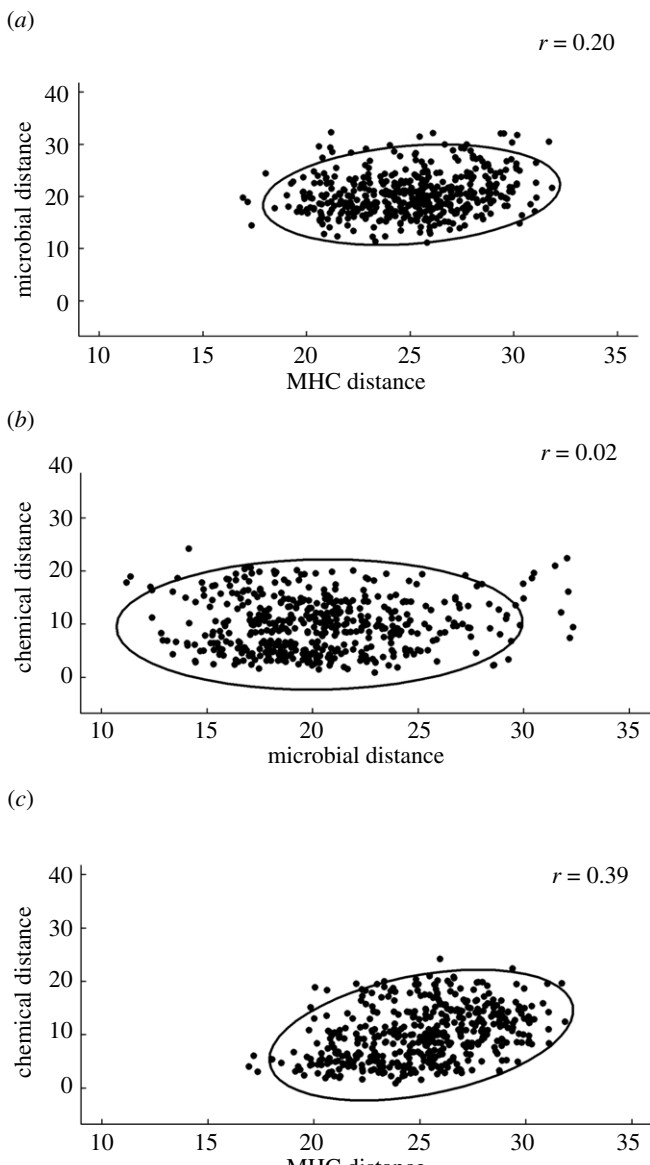

**Figure 1.** (*a*) Song sparrow dyads that were more similar at MHC class II also had more similar preen gland microbial communities. (*b*) Dyads with more similar preen gland microbial communities did not have more similar preen oil chemistry. (*c*) Dyads that were more similar at MHC class II had more similar preen oil chemistry. Euclidean distances were calculated from all pairwise dyads ($N = 31$ birds; 465 pairwise combinations). Ellipses indicate the normal-probability contours at 95% confidence.

## 4. Discussion

Understanding the potential interconnections between MHC, microbiota and animal odour is best addressed by examining all three links simultaneously [43]. Recent experimental evidence implicates preen gland microbes in producing behaviourally relevant preen oil compounds in dark-eyed juncos (*Junco hyemalis*) [24]. In wild and captive song sparrows [6,20] and black-legged kittiwakes (*Rissa tridactyla*) [44], individuals that are more similar at MHC class II also have more similar preen oil chemistry. However, a recent review [43] identifies a critical knowledge gap in that no studies have concurrently characterized MHC, microbiota and chemical profiles or the relationships between all three. To our knowledge, ours is the first study to do so.

We hypothesized that the correlation previously described between MHC class II genotype and preen oil chemistry [8,22,44] is mediated by microbial communities associated with the preen gland. That is, we hypothesized that MHC genotype shapes preen gland microbial communities, which in turn contribute to variation in the chemical composition of preen oil. Consistent with this microbial mediation hypothesis, dyads that were more similar at MHC also had more similar preen gland microbiota.

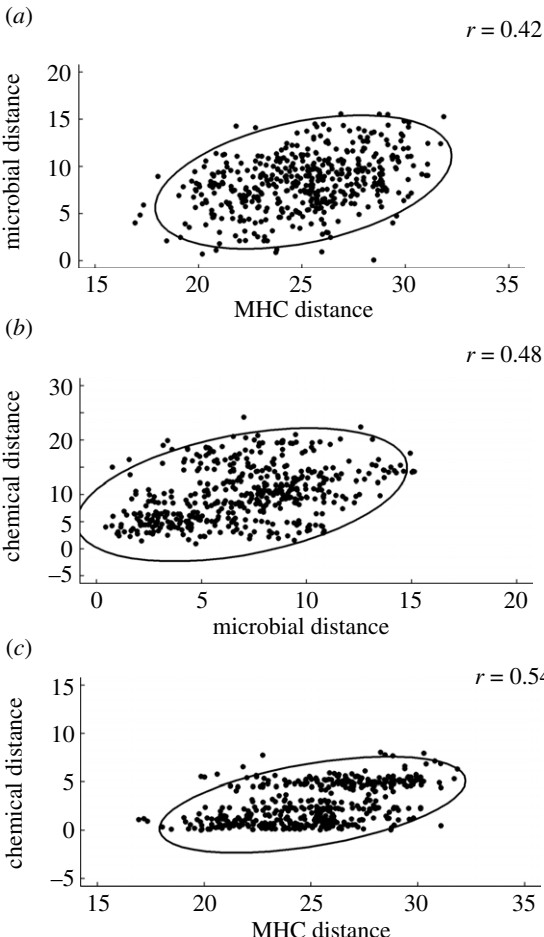

**Figure 2.** Permutation analyses identified subsets of up to six bacterial sequence variants that maximized the rank correlation between preen gland microbial distance and (*a*) MHC class II amino acid distance, and (*b*) preen oil chemical distance. (*c*) Permutation analysis identified a subset of six preen oil wax ester peaks that maximized the rank correlation between MHC class II amino acid distance and preen oil chemical distance. Euclidean distances were calculated from all pairwise dyads ($N =$ 31 birds; 465 pairwise combinations). Ellipses indicate the normal-probability contours at 95% confidence. Axis scales differ between panels because different subsets generate different distance ranges.

**Table 1.** Results of PERMANOVA using Euclidean distance matrices to test for population and sex differences in preen gland microbiota, MHC genotype and preen oil chemistry.

|  | d.f. | sum of squares | mean sum of squares | F | $R^2$ | *p*-value |
|---|---|---|---|---|---|---|
| microbes | | | | | | |
| population | 1 | 444.0 | 444.0 | 2.12 | 0.07 | 0.022 |
| sex | 1 | 121.1 | 121.1 | 0.58 | 0.02 | 0.926 |
| residuals | 28 | 5865.7 | 209.5 | — | 0.91 | — |
| MHC | | | | | | |
| population | 1 | 692.5 | 692.5 | 2.24 | 0.07 | 0.0002 |
| sex | 1 | 223.0 | 223.0 | 0.72 | 0.02 | 0.922 |
| residuals | 28 | 8640.0 | 308.6 | — | 0.90 | — |
| preen oil | | | | | | |
| population | 1 | 591.4 | 591.4 | 13.90 | 0.32 | <0.0001 |
| sex | 1 | 69.8 | 69.8 | 1.64 | 0.04 | 0.098 |
| residuals | 28 | 1191.1 | 42.5 | — | 0.64 | — |

However, inconsistent with the hypothesis, pairwise similarity in preen gland microbiota did not significantly predict similarity in the chemical composition of preen oil. Furthermore, when microbial similarity was calculated based on the full set of sequences retrieved from preen gland samples, the association between matrices of MHC and microbial distance and that between matrices of microbial and chemical distance were not consistently stronger than that between MHC and chemical distance. Together, these findings suggest that although preen gland microbial community composition appears sensitive to host genotype at MHC class II, the resulting variation in overall community composition does not drive individual variation in host chemical cues.

The fact that preen oil composition was more strongly related to MHC genotype than to overall preen gland microbiota ($r = 0.39$ versus $r = 0.02$) suggests that, counter to our prediction, the effects of MHC on preen oil composition are not mediated primarily through preen gland microbiota. Instead, MHC genotype may affect host odour more directly. Alternatively, the correlation we observed between MHC and preen oil chemical similarity could reflect indirect effects stemming from additional factors such as geographical variation [5,6].

MHC peptides bound to MHC proteins directly reflect the structure of the polymorphic peptide binding regions of MHC proteins. These MHC peptides can be secreted in bodily fluids, becoming chemical cues that convey information about MHC genotype [9,14,45]. This is consistent with our findings of a relatively large effect of MHC genotype on preen oil chemical composition, but a caveat is that we analysed the whole wax esters of preen oil rather than volatiles. We expect that such esters are the precursors to preen oil volatiles [25] but we did not measure volatile compounds directly, nor did we identify peptides or metabolites that might be MHC-derived. Consequently, our approach could have constrained our ability to detect effects of preen gland microbiota on odour.

Symbiotic microbes could affect preen oil composition during its production, storage and/or application. Bacteria living within the gland should be more likely to affect preen oil during production and/or storage, while bacteria living on the external surface of the gland and/or on the feathers should be more likely to affect preen oil after its excretion and application to the body surface. We sampled the overall bacterial community associated with the uropygial gland, so we are not able to reliably distinguish microbes from within and outside the gland. However, we reasoned that bacteria excreted with the preen oil we sampled immediately prior to swabbing and bacteria living on the external surface of the gland both have the potential to affect preen oil composition, since the oil is regularly excreted from the gland to the body surface.

The fermentation hypothesis for chemical recognition posits that symbiotic microbes produce host odours [46], but it is of course possible that not all gland-associated microbes do so. Similarly, some microbes may be more sensitive than others to host genotype at MHC class II. We supplemented our analyses of overall microbial distance with permutation analyses to identify subsets of microbes that maximize correlations between (i) microbial distance and MHC distance (candidates for being sensitive to host MHC genotype) or (ii) microbial distance and chemical distance (candidates for affecting host odour through modification of preen oil chemistry). The bacterial genera *Micrococcus* and *Bacillus* maximized correlations with MHC distance. *Micrococcus* and *Bacillus* have been found within the preen gland of dark-eyed juncos, and each contain species that are known odour producers [24]. For example, *M. luteus* breaks down sebaceous secretions such as human sweat, contributing to armpit odour [47]. Within the preen gland of birds, these genera may be able to degrade long-chain preen oil wax esters into odorous short-chain compounds. Our finding that the relative abundance of these genera varies with MHC genotype suggests that the metabolites these taxa produce may provide odour cues of MHC genotype.

We expected that the taxa identified through permutation analysis as maximizing correlations with preen oil chemical distance would include those that have previously been implicated in songbird chemical communication, such as *Pseudomonas*, *Burkholderia* or *Staphylococcus* [24,48]. We did not observe this pattern. Instead, genera *Methylobacterium*, *Bradyrhizobium* and *Xylophilus* maximized the rank correlation to preen oil chemistry. These taxa have primarily been found associated with plants and soils [49] and may thus be transiently acquired from the environment rather than being true preen gland symbionts. However, *Methylobacterium* and *Bradyrhizobium* have been found in or around the preen gland or on body feathers of other bird species [24,48,50–55].

Some of the genera we identified in association with the preen gland of song sparrows (e.g. *Bacillus*, *Methylobacterium* and *Sphingomonas*) can be found as contaminants in DNA extraction kits and laboratory reagents [56]. While we cannot be certain that no contaminant SVs remained in our final dataset, we

think it likely that the filtering steps we took removed contaminant SVs, as these would have been orders of magnitude less abundant than the bacteria recovered from the birds. Finally, although we hypothesized that microbial community structure influences preen gland chemistry (as has been shown previously [24]), we cannot rule out the alternative hypothesis that preen oil chemical composition influences the microbiome, by presenting a more or less favourable environment for certain species to inhabit.

The preen oil peaks that maximized correlations between preen oil chemical composition and MHC genotype were mixtures of $C12:C19$ through $C19:C16$ acid : alcohol monoesters. By contrast to previous results, these peaks do not match the preen oil peaks previously identified as best explaining the relationship between preen oil and MHC genetic similarity in song sparrows from Newboro, Ontario [22]. We sampled birds from London and Cambridge (approx. 400–500 km away from Newboro); thus, differences in influential preen oil peaks could reflect population differences in preen oil composition [25] and MHC genotype.

Our use of two populations may have contributed to the stronger relationship observed between MHC similarity and preen oil similarity ($r = 0.38$ for all pairwise dyads) relative to previous studies in this and other species (song sparrows, $r = 0.11–0.13$ [8,22]; black-legged kittiwakes ($r = 0.13–0.22$) [44]). Previous studies each focused on a single population, while we screened two geographically distinct populations that differed in the composition of MHC, microbes and preen oil (as inferred from greater pairwise distances between than within sites for each of these measures). Thus, we cannot exclude the possibility that significant correlations observed between MHC and both preen gland microbes and preen oil chemical composition do not exclusively reflect causal relationships but are driven at least partly by geographical differences. Consistent with this possibility, preen gland microbial communities are influenced by environmental factors, though genetic factors also play a role [20]. Another limitation of our study is that we are unable to disentangle the potential influence of geographical differences from the possibility that seasonal or physiological differences may have affected our results. Due to personnel limitations, we sampled the preen gland microbiota of London birds during post-breeding and of Cambridge birds during breeding. Seasonal changes in bird bacterial communities have not been well studied, but they have been observed in cloacal bacteria [57]. Surprisingly, we did not detect sex differences in preen oil chemical composition, despite collecting preen oil from breeding condition birds. This is in contrast to prior work in this species [25,26], and we suspect may be due to the smaller sample size used here.

# 5. Conclusion

Song sparrows with more similar MHC genotypes were more similar in the overall community structure of their preen gland microbiota. Overall microbial similarity did not predict chemical similarity of preen oil, despite a robust positive correlation between MHC similarity and chemical similarity, but permutation analysis identified a microbial subset that was strongly predictive of preen oil chemistry. Taken together, our findings suggest that previously reported correlations between MHC and preen oil composition may be explained partly by direct effects of MHC on preen oil chemical composition and partly by indirect effects mediated by a subset of microbes inhabiting the preen gland.

Ethics. All birds were captured under permission from the Canadian Wildlife Service and Environment and Climate Change Canada (Scientific Collection Permit CA 0244; banding subpermits 10691B, E, F). All procedures were approved by The University of Western Ontario Animal Use Subcommittee (protocol no. 2016-017 to E.A.M.-S.).

Data accessibility. The 16S rRNA gene sequencing files are available from the European Nucleotide Archive, study accession number PRJEB42688. MHC allele sequences are available on GenBank (accession numbers KX263957–KX375330; MF197789–MF197829; MK504126; and MH670972–MH671104 for 135 previously described sequences, accession numbers MZ592790-MZ592804 for 15 newly described sequences). Other supporting data are on the Dryad Digital Repository: https://dx.doi.org/10.5061/dryad.4f4qrfj9t.

Authors' contributions. L.A.G., G.B.G. and E.A.M.-S. designed the study. L.A.G. conducted field sampling and genetic analysis. L.A.G. conducted preen oil analysis in consultation with M.A.B. L.A.G. performed statistical analyses and drafted the manuscript in consultation with G.B.G. and E.A.M.-S. All authors gave approval for submission.

Competing interests. We declare we have no competing interests.

Funding. This study was supported by the Natural Sciences and Engineering Research Council of Canada (NSERC) through a Discovery Grant to E.A.M.-S. and a Vanier Canada Graduate Scholarship to L.A.G.

Acknowledgements. We thank four anonymous reviewers for comments that improved this manuscript and the *rare* Charitable Research Reserve for generously providing access to their land. This research was conducted on the traditional territories of the Anishnaabek, Attawanderin, Haudenoshaunee, Huron-Wendat, Lenape and Mississauga.

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
