## [Peer Review File · Royal Society Open Science]

Review History

RSOS-210936.R0 (Original submission)

Review form: Reviewer 1

Is the manuscript scientifically sound in its present form?

Yes

Are the interpretations and conclusions justified by the results?

Yes

Is the language acceptable?

Yes

Do you have any ethical concerns with this paper?

No

Have you any concerns about statistical analyses in this paper?

No

Recommendation?

Accept with minor revision (please list in comments)

Comments to the Author(s)

I now have read the manuscript "Preen gland microbiota covary with major histocompatibility complex genotype in a songbird" by Grieves et al.. The authors explore the hypothesis that odors, which most likely are used to determine MHC dissimilarity or variety, are produced by microbiota and should therefore correlate with MHC genotype. To do so, the authors collected preen gland secretion, microbiota and determined the MHC genotype of song sparrows (*Melospiza melodia*).

The authors found that, consistent with their hypothesis, pairwise similarity at the MHC correlated with microbiota collected around the preen gland. In contrast to the hypothesis however, overall microbial similarity did not correlate with chemical similarity. Nevertheless, using a permutation test they were able to find a subset of microbiota that correlate with preen oil similarity. It is still unknown how MHC diversity and similarity/distance is encoded in odors, although there is quite some evidence that odors are the main source to transfer information of the MHC genotype. This study by Grieves et al. uses a statistical approach to determine potential microbiota that might play a role.

Although this study is descriptive and correlative, I really enjoyed it and I think this is a good first step of disentangling the interplay between microbiota, odour and MHC.

The authors did a very good job in responding to the criticism that have been raised by the previous referees and as mentioned already earlier I am super happy with the manuscript and the interpretation of the results. I particularly like the statistical approach of finding subsets that are good predictors of preen oil composition and MHC genotype. All three types of data are compositional data, and therefore it is very likely that majority of microbiota are not involved in MHC signaling, making it important to determine the potential subsets that might play a role. With this approach it might be possible to perform experiments, when being able to manipulate the specific microbiota.

I have only a few minor comments.

line 51: There are a few examples where selection is favoring a optimum rather than a maximum. This possibility should be named.

line 69: If there are studies that have investigated this, name them. If not please say "this has, to our knowledge not been studied"

After reading the comments and response to the referees in the first round, I think this manuscript improved a lot and I recommend to add the two anonymous referees to the acknowledgements.

Last not least I wasn't able to check the data, as there was no link, but the authors made clear that the data will be available, once being published.

Review form: Reviewer 2**Is the manuscript scientifically sound in its present form?**

Yes

Are the interpretations and conclusions justified by the results?

Yes

Is the language acceptable?

Yes

Do you have any ethical concerns with this paper?

No

Have you any concerns about statistical analyses in this paper?

Yes

Recommendation?

Major revision is needed (please make suggestions in comments)

Comments to the Author(s)

Comments to the Authors:

I really enjoyed reading and reviewing this well written manuscript. The study is one of the first to concurrently examine the relationships between bacteria, MHC genes, and the chemical composition of bodily secretions. This work represents a much-needed step in the field of vertebrate chemical communication where the links between these three variables have long been discussed, but rarely (or never) explored all at once. I have very few minor comments as this manuscript is already very clear. However, I do have two major comments that concern the analysis approach, which I have described below.

Major Comments:

There seem to be a number of important ecological factors that are not accounted for in the analyses used in the MS. These factors include the sex of the individuals, the population of origin, and the breeding status. Based on previous findings documented in song sparrows (the focal species of this MS), and results from other bird species, each of these factors could impact preen oil chemistry and microbial communities, and the two populations could also differ in the distribution and abundance of MHC alleles. The results presented in the manuscript are convincing, so ignoring the potential effects of these ecological factors does not seem to be a major issue. However, I wonder whether certain patterns are lost. Most importantly, the justification behind the decision to pool all the samples and to not examine the effects of these factors on the data is missing from the MS. While I am not entirely opposed to the analysis approach that was used, I would like to see both a clear justification behind the decision to pool the birds and a better discussion of the caveats associated with this decision. I have provided some more specific thoughts and suggestions below broken up by each of the three factors (population, breeding status, sex):

Population: Lines 105-107 describe two populations that were sampled. Line 236 describes the analyses used to compare the preen oil chemistry, bacterial communities, and MHC alleles of the two populations. Lines 270-272 and Table 1 present the results of these analysis. I have a few comments:

- 1) The analyses and results that compare the two populations occur at the end of the methods and results, which makes them read like an after-thought. In their current location, this is confusing because it is unclear why this information is being provided. I suggest presenting this information earlier in both the methods and results, preferably before the Mantel tests and bioenv correlations.
- 2) The results of the t-tests in Table 1 suggest differences between the populations for MHC, microbes, and preen oil, but there is no rationale provided about why population-specific analyses were not performed. Is this a sample size issue? Would there be too little statistical power if looking at each population separately? The discussion (lines 361-371) does a good job of

explaining how the two populations may have influenced the results. I think a separate statement to explain why the two populations were pooled is needed much earlier in the MS, such as in the methods.

3) Why was a t-test used for this analysis? A permanova or anosim would be better suited to determining whether the microbes, MHC alleles, and preen oil chemistry differed between the two populations and the results would be easier to interpret.

Breeding status: Lines 105-107 also describe how the individuals used in this study were caught at different points in the year. The London birds were captured in August and September and the Cambridge birds in April and May. I imagine these sampling times correspond to different points in the breeding season (early and late) or even possibly to the non-breeding and breeding season. The chemical composition of preen oil chemicals changes with breeding status in song sparrows (Grieves et al. 2019) as well as many other birds (Reneerkens et al. 2007, Tuttle et al. 2014, Fischer et al. 2017, Whittaker et al. 2019). Furthermore, in a number of vertebrate species, the chemical composition of body secretions is only correlated with genetic markers at certain times of the year (Charpentier et al. 2008, Milinski et al. 2010, Potier et al. 2018, Grogan et al. 2019). Seasonal changes in bird bacterial communities have not been well studied, but they have been observed in cloacal bacteria (Escallón et al. 2019). As many birds undergo changes in the size of their preen gland during the breeding season, it seems plausible that the preen gland bacterial community may also experience changes. To me, the issue of breeding status seems to be a really important factor that is currently unmentioned. Unfortunately, the difference in breeding status cannot be separated from any differences associated with population. A justification of why the influence of breeding status was not analyzed is needed in the methods. The discussion could also benefit from addressing the potential influence of breeding state and how this cannot be disentangled from population-level influences in this study.

Sex: The study includes 19 males and 12 females, but no information is provided about the influence of sex on the three types of data (bacteria, preen oil, MHC diversity). Mantel tests can be broken up into same-sex and opposite sex dyads (see Grogan et al. 2019; Leclaire et al. 2012; Slade et al. 2016). This is one way to account for sex-specific trends. Previous findings from song sparrows (which I believe include some of the data used in this MS) showed mixed findings for sex-differences; males and females do not differ in bacterial communities (Grieves et al. 2021), but they do differ in their preen oil chemistry (Grieves et al. 2019). It is unclear if the diversity of MHC alleles for the individuals in this study differs between males and females, but this could easily be tested. If preen oil is the only component that shows sex-specific differences, then proceeding with males and females pooled could be a valid approach. If sex-differences are present for MHC diversity and preen oil, then performing Mantel tests for different sex dyads may be needed. Like the potential effects of population and breeding status, I think details to explain why sex-specific effects were not addressed are also needed.

I have a few comments that relate to the analysis described on Lines 220 to 222 (methods), Lines 262 to 269 (results), and Lines 372 to 381 (discussion). First, I am unsure that doing repeated bioenv analysis steps is an appropriate use of this method. Using one step to improve the correlation followed by another step to again improve the correlation seems problematic. Have you seen this approach before? And/or is this an established use of this method? Secondly, I am sort of perplexed by what these results actually mean. I am hoping you can help me wrap my head around this and also provide a bit more insight in the MS to help future readers. The first step in the bioenv process identified 3 bacterial SVs (Methylobacterium, Bradyrhizobium, and Xylophilus) that maximized the correlation with the preen oil chemicals. The second step identified 3 MHC class II alleles (SOSP-DAB*537, SOSP-DAB*424, SOSP-DAB*425) that maximize the correlation with the 3 bacteria SVs identified in the previous step. As expected, there are high levels of MHC polymorphism within the sampled birds, which is evident by the finding described on Line 188 of 151 unique alleles with an average of 16.23 alleles per bird. I wonder

then what the 3 alleles selected by the second bioenv step really tell us given that these alleles probably only occur in a few individuals. Is this analysis suggesting that individuals with any of these 3 alleles have differing abundances of the 3 bacterial SVs when compared with individuals who do not have these alleles? Moreover, it is unclear how common the 3 bacterial SVs are across the sampled birds. If both the 3 SVs and the 3 MHC alleles are rare, then I am really unclear on what information this analysis provides as it would seem to exclude a large number of the birds sampled. Additional clarification on how to interpret the results of this analysis would be useful.

Minor Comments:

Line 90 states that not all microbes contribute to host odor, which is part of the rationale behind using the bioenv analysis. I agree with this statement. However, I also think that not all chemicals in the preen oil are produced by bacteria or influenced by MHC, and therefore may be contributing noise to your data in a similar way as some of the bacterial SVs. Several papers support this idea (Leclaire et al. 2012, 2014; Stoffel et al. 2015). I think adding this idea to either the introduction or methods (around line 213), would more clearly justify the decision to use bioenv to identify the subset of chemicals than best reflect MHC distance.

Line 121-124. Was this blood sample also used for MHC genotyping? I don't think that the methods currently mention where the DNA for the MHC analysis came from.

Line 146. I think a clarifying statement here that explains what a sequence variant is could be beneficial for readers who are not microbiologists. Is this a genus? Species? What kind of biological information can be extracted?

Line 167 states that samples with fewer than 5000 reads were removed from the analysis, but on Line 168 the filtered dataset is described as containing 31 samples, which I assume match with the 31 birds that were sampled (Line 106). Does this mean that none of the samples were dropped from the analysis? Please clarify how many were removed, if any.

Line 281: A change of wording is needed here to more accurately reflect the data used in this study. The preen oil chemicals measured here are not "odours", which is pointed out later in the discussion (Lines 305-310). These chemicals are potential precursors to volatile odours. It would be more accurate to use the term "chemical profiles" or "chemotypes" to describe the chemical dataset.

References

- Charpentier, M. J. E., M. Boulet, and C. M. Drea (2008). Smelling right: The scent of male lemurs advertises genetic quality and relatedness. *Molecular Ecology* 17:3225–3233.
- Escallón, C., L. K. Belden, and I. T. Moore (2019). The Cloacal Microbiome Changes with the Breeding Season in a Wild Bird. *Integrative Organismal Biology* 1.
- Fischer, I., Ł. P. Haliński, W. Meissner, P. Stepnowski, and M. Knitter (2017). Seasonal changes in the preen wax composition of the Herring gull *Larus argentatus*. *Chemoecology* 27:127–139.
- Grieves, L. A., M. A. Bernards, and E. A. MacDougall-Shackleton (2019). Wax Ester Composition of Songbird Preen Oil Varies Seasonally and Differs between Sexes, Ages, and Populations. *Journal of Chemical Ecology* 45:37–45.
- Grieves, L. A., G. B. Gloor, T. R. Kelly, M. A. Bernards, and E. A. MacDougall-Shackleton (2021). Preen gland microbiota of songbirds differ across populations but not sexes. *Journal of Animal Ecology*. <https://doi.org/10.1111/1365-2656.13531>
- Grogan, K. E., R. L. Harris, M. Boulet, and C. M. Drea (2019). Genetic variation at MHC class II loci influences both olfactory signals and scent discrimination in ring-tailed lemurs. *BMC Evolutionary Biology* 19:171.

Leclaire, S., W. F. D. van Dongen, S. Voccia, T. Merklings, C. Ducamp, S. a Hatch, P. Blanchard, E. Danchin, R. H. Wagner, É. Danchin, and R. H. Wagner (2014). Preen secretions encode information on MHC similarity in certain sex-dyads in a monogamous seabird. *Scientific reports* 4:6920.

Leclaire, S., T. Merklings, C. Raynaud, H. Mulard, J.-M. Bessi re,  . Lhuillier, S. a Hatch, and  . Danchin (2012). Semiochemical compounds of preen secretion reflect genetic make-up in a seabird species. *Proceedings of the Royal Society B: Biological Sciences* 279:1185–1193.

Milinski, M., S. W. Griffiths, T. B. H. Reusch, and T. Boehm (2010). Costly major histocompatibility complex signals produced only by reproductively active males, but not females, must be validated by a “maleness signal” in three-spined sticklebacks. *Proceedings of the Royal Society B: Biological Sciences* 277:391–398.

Potier, S., M. M. Besnard, D. Schikorski, B. Buatois, O. Duriez, M. Gabirot, S. Leclaire, and F. Bonadonna (2018). Preen oil chemical composition encodes individuality, seasonal variation and kinship in black kites *Milvus migrans*. *Journal of Avian Biology* 49:e01728.

Reneerkens, J., T. Piersma, and J. S. Sinninghe Damste (2007). Expression of Annual Cycles in Preen Wax Composition in Red Knots: Constraints on the Changing Phenotype. *Journal of experimental zoology* 307A:127–139.

Slade, J. W. G., M. J. Watson, T. R. Kelly, G. B. Gloor, M. A. Bernards, and E. A. MacDougall-Shackleton (2016). Chemical composition of preen wax reflects major histocompatibility complex similarity in songbirds. *Proceedings of the Royal Society B: Biological Sciences* 283.

Stoffel, M. A., B. A. Caspers, J. Forcada, A. Giannakara, M. Baier, L. Eberhart-Phillips, C. M ller, and J. I. Hoffman (2015). Chemical fingerprints encode mother–offspring similarity, colony membership, relatedness, and genetic quality in fur seals. *Proceedings of the National Academy of Sciences* 112:E5005–E5012.

Tuttle, E. M., P. J. Sebastian, A. L. Posto, H. A. Soini, M. V Novotny, and R. A. Gonser (2014). Variation in preen oil composition pertaining to season, sex, and genotype in the polymorphic white-throated sparrow. *Journal of Chemical Ecology* 40:1025–1038.

Whittaker, D. J., M. Kuzel, M. J. E. Burrell, H. A. Soini, M. V. Novotny, and E. H. DuVal (2019). Chemical profiles reflect heterozygosity and seasonality in a tropical lekking passerine bird. *Animal Behaviour* 151:67–75.

Decision letter (RSOS-210936.R0)

Dear Dr Grieves

The Editors assigned to your paper RSOS-210936 "Preen gland microbiota covary with major histocompatibility complex genotype in a songbird" have now received comments from reviewers and would like you to revise the paper in accordance with the reviewer comments and any comments from the Editors. Please note this decision does not guarantee eventual acceptance.

Please submit your revised manuscript and required files (see below) no later than 21 days from today's (ie 09-Aug-2021) date. Note: the ScholarOne system will 'lock' if submission of the revision is attempted 21 or more days after the deadline. If you do not think you will be able to meet this deadline please contact the editorial office immediately.

on behalf of Dr Cynthia Downs (Associate Editor) and Kevin Padian (Subject Editor)
openscience@royalsociety.org

Associate Editor Comments to Author (Dr Cynthia Downs):
Comments to the Author:

Two expert reviewers and I have reviewed this manuscript. As summarized by Reviewer 1, the study found "...pairwise similarity at the MHC correlated with microbiota collected around the preen gland....[and] overall microbial similarity did not correlate with chemical similarity. Nevertheless, using a permutation test [the study was] able to find a subset of microbiota that correlate with preen oil similarity."

Both reviewers commented that they enjoyed reading the manuscript. They also noted that the study presented in the manuscript provides an essential link among microbiota, MHC genes, and the chemical composition of bodily secretions. The reviewers both point out a few minor edits, which I encourage the authors to address.

Reviewer 2 has a few major edits. The most important is the need for additional analyses around ecological factors, namely population, breeding status, and sex. Given the lack of an overall relationship between microbial similarity and chemical similarity, I encourage the authors to explore the role of ecological variables in their analyses. Not accounting for these factors may obscure important patterns relevant to the central hypothesis of the study. Note: I do not see a need to justify pooling the samples since samples from each microbiome were uniquely tagged.

Reviewer 2 also raises concerns about the statistical methods. Please clarify and justify the statistical methods used in this manuscript.

I enjoyed reading this manuscript and look forward to seeing a revision.

Reviewer comments to Author:

Reviewer: 1

Comments to the Author(s)

I now have read the manuscript "Preen gland microbiota covary with major histocompatibility complex genotype in a songbird" by Grieves et al.. The authors explore the hypothesis that odors, which most likely are used to determine MHC dissimilarity or variety, are produced by microbiota and should therefore correlate with MHC genotype. To do so, the authors collected preen gland secretion, microbiota and determined the MHC genotype of song sparrows (*Melospiza melodia*).

The authors found that, consistent with their hypothesis, pairwise similarity at the MHC correlated with microbiota collected around the preen gland. In contrast to the hypothesis however, overall microbial similarity did not correlate with chemical similarity. Nevertheless, using a permutation test they were able to find a subset of microbiota that correlate with preen oil similarity. It is still unknown how MHC diversity and similarity/distance is encoded in odors, although there is quite some evidence that odors are the main source to transfer information of the MHC genotype. This study by Grieves et al. uses a statistical approach to determine potential microbiota that might play a role.

Although this study is descriptive and correlative, I really enjoyed it and I think this is a good first step of disentangling the interplay between microbiota, odour and MHC.

The authors did a very good job in responding to the criticism that have been raised by the previous referees and as mentioned already earlier I am super happy with the manuscript and the interpretation of the results. I particularly like the statistical approach of finding subsets that are good predictors of preen oil composition and MHC genotype. All three types of data are compositional data, and therefore it is very likely that majority of microbiota are not involved in MHC signaling, making it important to determine the potential subsets that might play a role. With this approach it might be possible to perform experiments, when being able to manipulate the specific microbiota.

I have only a few minor comments.

line 51: There are a few examples where selection is favoring a optimum rather than a maximum. This possibility should be named.

line 69: If there are studies that have investigated this, name them. If not please say "this has, to our knowledge not been studied"

After reading the comments and response to the referees in the first round, I think this manuscript improved a lot and I recommend to add the two anonymous referees to the acknowledgements.

Last not least I wasn't able to check the data, as there was no link, but the authors made clear that the data will be available, once being published.

Reviewer: 2

Comments to the Author(s)

Comments to the Authors:

I really enjoyed reading and reviewing this well written manuscript. The study is one of the first to concurrently examine the relationships between bacteria, MHC genes, and the chemical composition of bodily secretions. This work represents a much-needed step in the field of vertebrate chemical communication where the links between these three variables have long been discussed, but rarely (or never) explored all at once. I have very few minor comments as this

manuscript is already very clear. However, I do have two major comments that concern the analysis approach, which I have described below.

Major Comments:

There seem to be a number of important ecological factors that are not accounted for in the analyses used in the MS. These factors include the sex of the individuals, the population of origin, and the breeding status. Based on previous findings documented in song sparrows (the focal species of this MS), and results from other bird species, each of these factors could impact preen oil chemistry and microbial communities, and the two populations could also differ in the distribution and abundance of MHC alleles. The results presented in the manuscript are convincing, so ignoring the potential effects of these ecological factors does not seem to be a major issue. However, I wonder whether certain patterns are lost. Most importantly, the justification behind the decision to pool all the samples and to not examine the effects of these factors on the data is missing from the MS. While I am not entirely opposed to the analysis approach that was used, I would like to see both a clear justification behind the decision to pool the birds and a better discussion of the caveats associated with this decision. I have provided some more specific thoughts and suggestions below broken up by each of the three factors (population, breeding status, sex):

Population: Lines 105-107 describe two populations that were sampled. Line 236 describes the analyses used to compare the preen oil chemistry, bacterial communities, and MHC alleles of the two populations. Lines 270-272 and Table 1 present the results of these analysis. I have a few comments:

- 1) The analyses and results that compare the two populations occur at the end of the methods and results, which makes them read like an after-thought. In their current location, this is confusing because it is unclear why this information is being provided. I suggest presenting this information earlier in both the methods and results, preferably before the Mantel tests and bioenv correlations.
- 2) The results of the t-tests in Table 1 suggest differences between the populations for MHC, microbes, and preen oil, but there is no rationale provided about why population-specific analyses were not performed. Is this a sample size issue? Would there be too little statistical power if looking at each population separately? The discussion (lines 361-371) does a good job of explaining how the two populations may have influenced the results. I think a separate statement to explain why the two populations were pooled is needed much earlier in the MS, such as in the methods.
- 3) Why was a t-test used for this analysis? A permanova or anosim would be better suited to determining whether the microbes, MHC alleles, and preen oil chemistry differed between the two populations and the results would be easier to interpret.

Breeding status: Lines 105-107 also describe how the individuals used in this study were caught at different points in the year. The London birds were captured in August and September and the Cambridge birds in April and May. I imagine these sampling times correspond to different points in the breeding season (early and late) or even possibly to the non-breeding and breeding season. The chemical composition of preen oil chemicals changes with breeding status in song sparrows (Grieves et al. 2019) as well as many other birds (Reneerkens et al. 2007, Tuttle et al. 2014, Fischer et al. 2017, Whittaker et al. 2019). Furthermore, in a number of vertebrate species, the chemical composition of body secretions is only correlated with genetic markers at certain times of the year (Charpentier et al. 2008, Milinski et al. 2010, Potier et al. 2018, Grogan et al. 2019). Seasonal changes in bird bacterial communities have not been well studied, but they have been observed in cloacal bacteria (Escallón et al. 2019). As many birds undergo changes in the size of their preen gland during the breeding season, it seems plausible that the preen gland bacterial community may also experience changes. To me, the issue of breeding status seems to be a really important

factor that is currently unmentioned. Unfortunately, the difference in breeding status cannot be separated from any differences associated with population. A justification of why the influence of breeding status was not analyzed is needed in the methods. The discussion could also benefit from addressing the potential influence of breeding state and how this cannot be disentangled from population-level influences in this study.

Sex: The study includes 19 males and 12 females, but no information is provided about the influence of sex on the three types of data (bacteria, preen oil, MHC diversity). Mantel tests can be broken up into same-sex and opposite sex dyads (see Grogan et al. 2019; Leclaire et al. 2012; Slade et al. 2016). This is one way to account for sex-specific trends. Previous findings from song sparrows (which I believe include some of the data used in this MS) showed mixed findings for sex-differences; males and females do not differ in bacterial communities (Grieves et al. 2021), but they do differ in their preen oil chemistry (Grieves et al. 2019). It is unclear if the diversity of MHC alleles for the individuals in this study differs between males and females, but this could easily be tested. If preen oil is the only component that shows sex-specific differences, then proceeding with males and females pooled could be a valid approach. If sex-differences are present for MHC diversity and preen oil, then performing Mantel tests for different sex dyads may be needed. Like the potential effects of population and breeding status, I think details to explain why sex-specific effects were not addressed are also needed.

I have a few comments that relate to the analysis described on Lines 220 to 222 (methods), Lines 262 to 269 (results), and Lines 372 to 381 (discussion). First, I am unsure that doing repeated bioenv analysis steps is an appropriate use of this method. Using one step to improve the correlation followed by another step to again improve the correlation seems problematic. Have you seen this approach before? And/or is this an established use of this method? Secondly, I am sort of perplexed by what these results actually mean. I am hoping you can help me wrap my head around this and also provide a bit more insight in the MS to help future readers. The first step in the bioenv process identified 3 bacterial SVs (*Methylobacterium*, *Bradyrhizobium*, and *Xylophilus*) that maximized the correlation with the preen oil chemicals. The second step identified 3 MHC class II alleles (SOSP-DAB*537, SOSP-DAB*424, SOSP-DAB*425) that maximize the correlation with the 3 bacteria SVs identified in the previous step. As expected, there are high levels of MHC polymorphism within the sampled birds, which is evident by the finding described on Line 188 of 151 unique alleles with an average of 16.23 alleles per bird. I wonder then what the 3 alleles selected by the second bioenv step really tell us given that these alleles probably only occur in a few individuals. Is this analysis suggesting that individuals with any of these 3 alleles have differing abundances of the 3 bacterial SVs when compared with individuals who do not have these alleles? Moreover, it is unclear how common the 3 bacterial SVs are across the sampled birds. If both the 3 SVs and the 3 MHC alleles are rare, then I am really unclear on what information this analysis provides as it would seem to exclude a large number of the birds sampled. Additional clarification on how to interpret the results of this analysis would be useful.

Minor Comments:

Line 90 states that not all microbes contribute to host odor, which is part of the rationale behind using the bioenv analysis. I agree with this statement. However, I also think that not all chemicals in the preen oil are produced by bacteria or influenced by MHC, and therefore may be contributing noise to your data in a similar way as some of the bacterial SVs. Several papers support this idea (Leclaire et al. 2012, 2014; Stoffel et al. 2015). I think adding this idea to either the introduction or methods (around line 213), would more clearly justify the decision to use bioenv to identify the subset of chemicals that best reflect MHC distance.

Line 121-124. Was this blood sample also used for MHC genotyping? I don't think that the methods currently mention where the DNA for the MHC analysis came from.

Line 146. I think a clarifying statement here that explains what a sequence variant is could be beneficial for readers who are not microbiologists. Is this a genus? Species? What kind of biological information can be extracted?

Line 167 states that samples with fewer than 5000 reads were removed from the analysis, but on Line 168 the filtered dataset is described as containing 31 samples, which I assume match with the 31 birds that were sampled (Line 106). Does this mean that none of the samples were dropped from the analysis? Please clarify how many were removed, if any.

Line 281: A change of wording is needed here to more accurately reflect the data used in this study. The preen oil chemicals measured here are not “odours”, which is pointed out later in the discussion (Lines 305-310). These chemicals are potential precursors to volatile odours. It would be more accurate to use the term “chemical profiles” or “chemotypes” to describe the chemical dataset.

References

- Charpentier, M. J. E., M. Boulet, and C. M. Drea (2008). Smelling right: The scent of male lemurs advertises genetic quality and relatedness. *Molecular Ecology* 17:3225–3233.
- Escallón, C., L. K. Belden, and I. T. Moore (2019). The Cloacal Microbiome Changes with the Breeding Season in a Wild Bird. *Integrative Organismal Biology* 1.
- Fischer, I., L. P. Haliński, W. Meissner, P. Stepnowski, and M. Knitter (2017). Seasonal changes in the preen wax composition of the Herring gull *Larus argentatus*. *Chemoecology* 27:127–139.
- Grieves, L. A., M. A. Bernards, and E. A. MacDougall-Shackleton (2019). Wax Ester Composition of Songbird Preen Oil Varies Seasonally and Differs between Sexes, Ages, and Populations. *Journal of Chemical Ecology* 45:37–45.
- Grieves, L. A., G. B. Gloor, T. R. Kelly, M. A. Bernards, and E. A. MacDougall-Shackleton (2021). Preen gland microbiota of songbirds differ across populations but not sexes. *Journal of Animal Ecology*. <https://doi.org/10.1111/1365-2656.13531>
- Grogan, K. E., R. L. Harris, M. Boulet, and C. M. Drea (2019). Genetic variation at MHC class II loci influences both olfactory signals and scent discrimination in ring-tailed lemurs. *BMC Evolutionary Biology* 19:171.
- Leclaire, S., W. F. D. van Dongen, S. Voccia, T. Merkling, C. Ducamp, S. a Hatch, P. Blanchard, E. Danchin, R. H. Wagner, É. Danchin, and R. H. Wagner (2014). Preen secretions encode information on MHC similarity in certain sex-dyads in a monogamous seabird. *Scientific reports* 4:6920.
- Leclaire, S., T. Merkling, C. Raynaud, H. Mulard, J.-M. Bessière, É. Lhuillier, S. a Hatch, and É. Danchin (2012). Semiochemical compounds of preen secretion reflect genetic make-up in a seabird species. *Proceedings of the Royal Society B: Biological Sciences* 279:1185–1193.
- Milinski, M., S. W. Griffiths, T. B. H. Reusch, and T. Boehm (2010). Costly major histocompatibility complex signals produced only by reproductively active males, but not females, must be validated by a “maleness signal” in three-spined sticklebacks. *Proceedings of the Royal Society B: Biological Sciences* 277:391–398.
- Potier, S., M. M. Besnard, D. Schikorski, B. Buatois, O. Duriez, M. Gabirot, S. Leclaire, and F. Bonadonna (2018). Preen oil chemical composition encodes individuality, seasonal variation and kinship in black kites *Milvus migrans*. *Journal of Avian Biology* 49:e01728.
- Reneerkens, J., T. Piersma, and J. S. Sinninghe Damste (2007). Expression of Annual Cycles in Preen Wax Composition in Red Knots: Constraints on the Changing Phenotype. *Journal of experimental zoology* 307A:127–139.
- Slade, J. W. G., M. J. Watson, T. R. Kelly, G. B. Gloor, M. A. Bernards, and E. A. MacDougall-Shackleton (2016). Chemical composition of preen wax reflects major histocompatibility complex similarity in songbirds. *Proceedings of the Royal Society B: Biological Sciences* 283.

Stoffel, M. A., B. A. Caspers, J. Forcada, A. Giannakara, M. Baier, L. Eberhart-Phillips, C. Müller, and J. I. Hoffman (2015). Chemical fingerprints encode mother–offspring similarity, colony membership, relatedness, and genetic quality in fur seals. *Proceedings of the National Academy of Sciences* 112:E5005–E5012.

Tuttle, E. M., P. J. Sebastian, A. L. Posto, H. A. Soini, M. V Novotny, and R. A. Gonser (2014). Variation in preen oil composition pertaining to season, sex, and genotype in the polymorphic white-throated sparrow. *Journal of Chemical Ecology* 40:1025–1038.

Whittaker, D. J., M. Kuzel, M. J. E. Burrell, H. A. Soini, M. V. Novotny, and E. H. DuVal (2019). Chemical profiles reflect heterozygosity and seasonality in a tropical lekking passerine bird. *Animal Behaviour* 151:67–75.

===PREPARING YOUR MANUSCRIPT===

===PREPARING YOUR REVISION IN SCHOLARONE===

Author's Response to Decision Letter for (RSOS-210936.R0)

See Appendix A.

Decision letter (RSOS-210936.R1)

Dear Dr Grieves,

It is a pleasure to accept your manuscript entitled "Preen gland microbiota covary with major histocompatibility complex genotype in a songbird" in its current form for publication in Royal Society Open Science.

Please ensure that you send to the editorial office an editable version of your accepted manuscript, and individual files for each figure and table included in your manuscript. You can send these in a zip folder if more convenient. Failure to provide these files may delay the processing of your proof.

on behalf of Dr Cynthia Downs (Associate Editor) and Kevin Padian (Subject Editor)
openscience@royalsociety.org

Associate Editor Comments to Author (Dr Cynthia Downs):
Associate Editor
Comments to the Author:

I enjoyed reading the revisions of this manuscript. The manuscript added some intellectual interest to my Saturday afternoon. This version adequately addresses all of the reviewer comments. The study is scientifically sound, the results are interpreted within appropriate boundaries, and the work provides important links among the microbiome, odors, and the immune system. I have no additional suggestions.

Appendix A

2 September 2021

Dear Dr. Downs,

Thank you for the opportunity to revise and resubmit our manuscript, "Preen gland microbiota covary with major histocompatibility complex genotype in a songbird" (RSOS-210936). We have addressed all of the reviewer comments and our responses to are below in *italics*. We look forward to your final decision on our manuscript.

Sincerely,

Dr. Leanne Grieves

(on behalf of all co-authors)

Associate Editor Comments to Author (Dr Cynthia Downs):

Comments to the Author:

Two expert reviewers and I have reviewed this manuscript. As summarized by Reviewer 1, the study found "...pairwise similarity at the MHC correlated with microbiota collected around the preen gland....[and] overall microbial similarity did not correlate with chemical similarity. Nevertheless, using a permutation test [the study was] able to find a subset of microbiota that correlate with preen oil similarity."

Both reviewers commented that they enjoyed reading the manuscript. They also noted that the study presented in the manuscript provides an essential link among microbiota, MHC genes, and the chemical composition of bodily secretions. The reviewers both point out a few minor edits, which I encourage the authors to address.

Thank you for sharing this positive feedback.

Reviewer 2 has a few major edits. The most important is the need for additional analyses around ecological factors, namely population, breeding status, and sex. Given the lack of an overall relationship between microbial similarity and chemical similarity, I encourage the authors to explore the role of ecological variables in their analyses. Not accounting for these factors may obscure important patterns relevant to the central hypothesis of the study. Note: I do not see a need to justify pooling the samples since samples from each microbiome were uniquely tagged.

Reviewer 2 also raises concerns about the statistical methods. Please clarify and justify the statistical methods used in this manuscript.

I enjoyed reading this manuscript and look forward to seeing a revision.

Thank you. We have addressed Reviewer 2's comments as outlined above and in their detailed remarks.

Reviewer comments to Author:

Reviewer: 1

Comments to the Author(s)

I now have read the manuscript "Preen gland microbiota covary with major histocompatibility complex genotype in a songbird" by Grieves et al.. The authors explore the hypothesis that odors, which most likely are used to determine MHC dissimilarity or variety, are produced by microbiota and should therefore correlate with MHC genotype. To

do so , the authors collected preen gland secretion, microbiota and determined the MHC genotype of song sparrows (*Melospiza melodia*).

The authors found that, consistent with their hypothesis, pairwise similarity at the MHC correlated with microbiota collected around the preen gland. In contrast to the hypothesis however, overall microbial similarity did not correlate with chemical similarity. Nevertheless, using a permutation test they were able to find a subset of microbiota that correlate with preen oil similarity. It is still unknown how MHC diversity and similarity/distance is encoded in odors, although there is quite some evidence that odors are the main source to transfer information of the MHC genotype. This study by Grieves et al. uses a statistical approach to determine potential microbiota that might play a role. Although this study is descriptive and correlative, I really enjoyed it and I think this is a good first step of disentangling the interplay between microbiota, odour and MHC.

The authors did a very good job in responding to the criticism that have been raised by the previous referees and as mentioned already earlier I am super happy with the manuscript and the interpretation of the results. I particularly like the statistical approach of finding subsets that are good predictors of preen oil composition and MHC genotype. All three types of data are compositional data, and therefore it is very likely that majority of microbiota are not involved in MHC signaling, making it important to determine the potential subsets that might play a role. With this approach it might be possible to perform experiments, when being able to manipulate the specific microbiota.

Thank you for sharing this positive feedback.

I have only a few minor comments.

line 51: There are a few examples where selection is favoring a optimum rather than a maximum. This possibility should be named.

We have added this information on lines 49 – 52.

line 69: If there are studies that have investigated this , name them. If not please say "this has , to our knowledge not been studied"

Thank you for this comment. Two studies that have investigated effects of MHC genotype on avian microbiota are named in the next sentence.

“While the influence of MHC genotype on host microbial communities has received growing attention in recent years [16–18], this has rarely been explored in birds. However, in seabirds, MHC class II genotype has recently been correlated with microbiota of the feathers [19] and preen gland [20].”

After reading the comments and response to the referees in the first round, I think this manuscript improved a lot and I recommend to add the two anonymous referees to the acknowledgements.

Thank you for the reminder to do this. We have now added this to the acknowledgments, along with acknowledgment of the present reviewers' helpful feedback.

Last not least I wasn't able to check the data, as there was no link, but the authors made clear that the data will be available, once being published.

Thank you. The data are currently on embargo in Dryad and will be made available as soon as there is a doi available.

Reviewer: 2

Comments to the Author(s)

Comments to the Authors:

I really enjoyed reading and reviewing this well written manuscript. The study is one of the first to concurrently examine the relationships between bacteria, MHC genes, and the chemical composition of bodily secretions. This work represents a much-needed step in the field of vertebrate chemical communication where the links between these three variables have long been discussed, but rarely (or never) explored all at once. I have very few minor comments as this manuscript is already very clear. However, I do have two major comments that concern the analysis approach, which I have described below.

Thank you!

Major Comments:

There seem to be a number of important ecological factors that are not accounted for in the analyses used in the MS. These factors include the sex of the individuals, the population of origin, and the breeding status. Based on previous findings documented in song sparrows (the focal species of this MS), and results from other bird species, each of these factors could impact preen oil chemistry and microbial communities, and the two populations could also differ in the distribution and abundance of MHC alleles. The results presented in the manuscript are convincing, so ignoring the potential effects of these ecological factors does not seem to be a major issue. However, I wonder whether certain patterns are lost. Most importantly, the justification behind the decision to pool all the samples and to not examine the effects of these factors on the data is missing from the MS. While I am not entirely opposed to the analysis approach that was used, I would like to see both a clear justification behind the decision to pool the birds and a better discussion of the caveats associated with

this decision. I have provided some more specific thoughts and suggestions below broken up by each of the three factors (population, breeding status, sex):

Population: Lines 105-107 describe two populations that were sampled. Line 236 describes the analyses used to compare the preen oil chemistry, bacterial communities, and MHC alleles of the two populations. Lines 270-272 and Table 1 present the results of these analysis. I have a few comments:

1) The analyses and results that compare the two populations occur at the end of the methods and results, which makes them read like an after-thought. In their current location, this is confusing because it is unclear why this information is being provided. I suggest presenting this information earlier in both the methods and results, preferably before the Mantel tests and bioenv correlations.

We have moved this information to lines 219 – 227 of Methods and lines 261 – 263 of Results.

2) The results of the t-tests in Table 1 suggest differences between the populations for MHC, microbes, and preen oil, but there is no rationale provided about why population-specific analyses were not performed. Is this a sample size issue? Would there be too little statistical power if looking at each population separately? The discussion (lines 361-371) does a good job of explaining how the two populations may have influenced the results. I think a separate statement to explain why the two populations were pooled is needed much earlier in the MS, such as in the methods.

This was primarily a sample size/statistical power issue. We add justification in the methods on lines 224 – 227.

3) Why was a t-test used for this analysis? A permanova or anosim would be better suited to determining whether the microbes, MHC alleles, and preen oil chemistry differed between the two populations and the results would be easier to interpret.

Following this suggestion, we have replaced the t-tests with PERMANOVA (methods lines 219 – 227; results lines 261 – 263, discussion lines 369 – 379).

Breeding status: Lines 105-107 also describe how the individuals used in this study were caught at different points in the year. The London birds were captured in August and September and the Cambridge birds in April and May. I imagine these sampling times correspond to different points in the breeding season (early and late) or even possibly to the non-breeding and breeding season. The chemical composition of preen oil chemicals changes with breeding status in song sparrows (Grieves et al. 2019) as well as many other birds (Reneerkens et al. 2007, Tuttle et al. 2014, Fischer et al. 2017, Whittaker et al. 2019). Furthermore, in a number of vertebrate species, the chemical composition of body

secretions is only correlated with genetic markers at certain times of the year (Charpentier et al. 2008, Milinski et al. 2010, Potier et al. 2018, Grogan et al. 2019). Seasonal changes in bird bacterial communities have not been well studied, but they have been observed in cloacal bacteria (Escallón et al. 2019). As many birds undergo changes in the size of their preen gland during the breeding season, it seems plausible that the preen gland bacterial community may also experience changes. To me, the issue of breeding status seems to be a really important factor that is currently unmentioned. Unfortunately, the difference in breeding status cannot be separated from any differences associated with population. A justification of why the influence of breeding status was not analyzed is needed in the methods. The discussion could also benefit from addressing the potential influence of breeding state and how this cannot be disentangled from population-level influences in this study.

We add justification in the methods on lines 222 – 227. We add discussion to acknowledge the potential influence of reproductive condition on lines 379 – 387.

Sex: The study includes 19 males and 12 females, but no information is provided about the influence of sex on the three types of data (bacteria, preen oil, MHC diversity). Mantel tests can be broken up into same-sex and opposite sex dyads (see Grogan et al. 2019; Leclaire et al. 2012; Slade et al. 2016). This is one way to account for sex-specific trends. Previous findings from song sparrows (which I believe include some of the data used in this MS) showed mixed findings for sex-differences; males and females do not differ in bacterial communities (Grieves et al. 2021), but they do differ in their preen oil chemistry (Grieves et al. 2019). It is unclear if the diversity of MHC alleles for the individuals in this study differs between males and females, but this could easily be tested. If preen oil is the only component that shows sex-specific differences, then proceeding with males and females pooled could be a valid approach. If sex-differences are present for MHC diversity and preen oil, then performing Mantel tests for different sex dyads may be needed. Like the potential effects of population and breeding status, I think details to explain why sex-specific effects were not addressed are also needed.

Following this comment and point 3) above, we now include PERMANOVA tests for effects of site (population) and sex that replace the t-tests we originally performed. Similar to the t-test results, we find an effect of site but not sex on preen gland microbes, mhc genotype, and preen oil chemical composition. We present this information on lines 219 – 227 (methods); 261 – 263 (results), and 369 – 387 (discussion).

I have a few comments that relate to the analysis described on Lines 220 to 222 (methods), Lines 262 to 269 (results), and Lines 372 to 381 (discussion). First, I am unsure that doing repeated bioenv analysis steps is an appropriate use of this method. Using one step to improve the correlation followed by another step to again improve the correlation seems

problematic. Have you seen this approach before? And/or is this an established use of this method? Secondly, I am sort of perplexed by what these results actually mean. I am hoping you can help me wrap my head around this and also provide a bit more insight in the MS to help future readers. The first step in the bioenv process identified 3 bacterial SVs (Methylobacterium, Bradyrhizobium, and Xylophilus) that maximized the correlation with the preen oil chemicals. The second step identified 3 MHC class II alleles (SOSP-DAB*537, SOSP-DAB*424, SOSP-DAB*425) that maximize the correlation with the 3 bacteria SVs identified in the previous step. As expected, there are high levels of MHC polymorphism within the sampled birds, which is evident by the finding described on Line 188 of 151 unique alleles with an average of 16.23 alleles per bird. I wonder then what the 3 alleles selected by the second bioenv step really tell us given that these alleles probably only occur in a few individuals. Is this analysis suggesting that individuals with any of these 3 alleles have differing abundances of the 3 bacterial SVs when compared with individuals who do not have these alleles? Moreover, it is unclear how common the 3 bacterial SVs are across the sampled birds. If both the 3 SVs and the 3 MHC alleles are rare, then I am really unclear on what information this analysis provides as it would seem to exclude a large number of the birds sampled. Additional clarification on how to interpret the results of this analysis would be useful.

We agree with the reviewer's concerns about this method, which we were originally suggested to undertake by an earlier reviewer. However, we are not aware of other examples of bioenv being used in this way. Given this and the important points the present reviewer brings up above, we have decided to remove this analysis. If the reviewer would prefer that we retain the analysis but add additional information and clarification, we will be happy to do so.

Minor Comments:

Line 90 states that not all microbes contribute to host odor, which is part of the rationale behind using the bioenv analysis. I agree with this statement. However, I also think that not all chemicals in the preen oil are produced by bacteria or influenced by MHC, and therefore may be contributing noise to your data in a similar way as some of the bacterial SVs. Several papers support this idea (Leclaire et al. 2012, 2014; Stoffel et al. 2015). I think adding this idea to either the introduction or methods (around line 213), would more clearly justify the decision to use bioenv to identify the subset of chemicals that best reflect MHC distance. *We have added this to lines 228 – 229, and additional clarification to lines 230 – 235.*

Line 121-124. Was this blood sample also used for MHC genotyping? I don't think that the methods currently mention where the DNA for the MHC analysis came from. *Yes, it was. We now state this on line 126.*

Line 146. I think a clarifying statement here that explains what a sequence variant is could be beneficial for readers who are not microbiologists. Is this a genus? Species? What kind of biological information can be extracted?

Thank you for catching this issue. In this case, sequence variant refers to unique sequences. We have clarified on lines 149 – 150, and add additional detail on the level of taxonomic identification on line 173.

Line 167 states that samples with fewer than 5000 reads were removed from the analysis, but on Line 168 the filtered dataset is described as containing 31 samples, which I assume match with the 31 birds that were sampled (Line 106). Does this mean that none of the samples were dropped from the analysis? Please clarify how many were removed, if any.

Thank you for pointing this out. We used the same data processing methods for the present study and a related paper (Grieves et al. 2021 J Anim Ecol). The filtering step resulted in removal of some sequences from the larger data set, but not from the dataset used in this paper. We have now clarified this on lines 174 – 175.

Line 281: A change of wording is needed here to more accurately reflect the data used in this study. The preen oil chemicals measured here are not “odours”, which is pointed out later in the discussion (Lines 305-310). These chemicals are potential precursors to volatile odours. It would be more accurate to use the term “chemical profiles” or “chemotypes” to describe the chemical dataset.

We have changed this to ‘chemical profiles’ (Line 289).

References

- Charpentier, M. J. E., M. Boulet, and C. M. Drea (2008). Smelling right: The scent of male lemurs advertises genetic quality and relatedness. *Molecular Ecology* 17:3225–3233.
- Escallón, C., L. K. Belden, and I. T. Moore (2019). The Cloacal Microbiome Changes with the Breeding Season in a Wild Bird. *Integrative Organismal Biology* 1.
- Fischer, I., Ł. P. Haliński, W. Meissner, P. Stepnowski, and M. Knitter (2017). Seasonal changes in the preen wax composition of the Herring gull *Larus argentatus*. *Chemoecology* 27:127–139.
- Grieves, L. A., M. A. Bernards, and E. A. MacDougall-Shackleton (2019). Wax Ester Composition of Songbird Preen Oil Varies Seasonally and Differs between Sexes, Ages, and Populations. *Journal of Chemical Ecology* 45:37–45.
- Grieves, L. A., G. B. Gloor, T. R. Kelly, M. A. Bernards, and E. A. MacDougall-Shackleton (2021). Preen gland microbiota of songbirds differ across populations but not sexes. *Journal of Animal Ecology*. <https://doi.org/10.1111/1365-2656.13531>
- Grogan, K. E., R. L. Harris, M. Boulet, and C. M. Drea (2019). Genetic variation at MHC class II

loci influences both olfactory signals and scent discrimination in ring-tailed lemurs. *BMC Evolutionary Biology* 19:171.

Leclaire, S., W. F. D. van Dongen, S. Voccia, T. Merklings, C. Ducamp, S. a Hatch, P. Blanchard, E. Danchin, R. H. Wagner, É. Danchin, and R. H. Wagner (2014). Preen secretions encode information on MHC similarity in certain sex-dyads in a monogamous seabird. *Scientific reports* 4:6920.

Leclaire, S., T. Merklings, C. Raynaud, H. Mulard, J.-M. Bessi re,  . Lhuillier, S. a Hatch, and  . Danchin (2012). Semiochemical compounds of preen secretion reflect genetic make-up in a seabird species. *Proceedings of the Royal Society B: Biological Sciences* 279:1185–1193.

Milinski, M., S. W. Griffiths, T. B. H. Reusch, and T. Boehm (2010). Costly major histocompatibility complex signals produced only by reproductively active males, but not females, must be validated by a “maleness signal” in three-spined sticklebacks. *Proceedings of the Royal Society B: Biological Sciences* 277:391–398.

Potier, S., M. M. Besnard, D. Schikorski, B. Buatois, O. Duriez, M. Gabirot, S. Leclaire, and F. Bonadonna (2018). Preen oil chemical composition encodes individuality, seasonal variation and kinship in black kites *Milvus migrans*. *Journal of Avian Biology* 49:e01728.

Reneerkens, J., T. Piersma, and J. S. Sinninghe Damste (2007). Expression of Annual Cycles in Preen Wax Composition in Red Knots: Constraints on the Changing Phenotype. *Journal of experimental zoology* 307A:127–139.

Slade, J. W. G., M. J. Watson, T. R. Kelly, G. B. Gloor, M. A. Bernards, and E. A. MacDougall-Shackleton (2016). Chemical composition of preen wax reflects major histocompatibility complex similarity in songbirds. *Proceedings of the Royal Society B: Biological Sciences* 283.

Stoffel, M. A., B. A. Caspers, J. Forcada, A. Giannakara, M. Baier, L. Eberhart-Phillips, C. M ller, and J. I. Hoffman (2015). Chemical fingerprints encode mother–offspring similarity, colony membership, relatedness, and genetic quality in fur seals. *Proceedings of the National Academy of Sciences* 112:E5005–E5012.

Tuttle, E. M., P. J. Sebastian, A. L. Posto, H. A. Soini, M. V Novotny, and R. A. Gonser (2014). Variation in preen oil composition pertaining to season, sex, and genotype in the polymorphic white-throated sparrow. *Journal of Chemical Ecology* 40:1025–1038.

Whittaker, D. J., M. Kuzel, M. J. E. Burrell, H. A. Soini, M. V. Novotny, and E. H. DuVal (2019). Chemical profiles reflect heterozygosity and seasonality in a tropical lekking passerine bird. *Animal Behaviour* 151:67–75.

Thank you for providing these references to go along with your comments!